# Does Cannabidiol (CBD) in Food Supplements Pose a Serious Health Risk? Consequences of the European Food Safety Authority (EFSA) Clock Stop Regarding Novel Food Authorisation

**Dirk W. Lachenmeier \*** , **Constanze Sproll and Stephan G. Walch**

Chemisches und Veterinäruntersuchungsamt (CVUA) Karlsruhe, Weissenburger Strasse 3,
76187 Karlsruhe, Germany; constanze.sproll@cvuaka.bwl.de (C.S.); stephan.walch@cvuaka.bwl.de (S.G.W.)
**\*** Correspondence: lachenmeier@web.de; Tel.: +49-721-926-5434

**Abstract:** In the European Union (EU), cannabidiol (CBD) products extracted from *Cannabis sativa* L. require pre-marketing authorisation under the novel food regulation. Currently, 19 CBD applications are being assessed by the European Food Safety Authority (EFSA). During the initial assessment of the dossiers, the EFSA Panel on Nutrition, Novel Foods, and Food Allergens (NDA) identified several knowledge gaps that need to be addressed before the evaluation of CBD can be finalised. The effects of CBD on the liver, gastrointestinal tract, endocrine system, nervous system, psychological function, and reproductive system need to be clarified. The contribution of this research is to provide an evidence-based assessment of the potential risks associated with CBD products, and to provide recommendations for risk management in the European Union while awaiting the finalisation of novel food applications. The available literature allows for a benchmark dose (BMD)–response modelling of several bioassays, resulting in a BMD lower confidence limit (BMDL) of 20 mg/kg bw/day for liver toxicity in rats. Human data in healthy volunteers showed increases in the liver enzymes alanine aminotransferase and aspartate aminotransferase in one study at 4.3 mg/kg bw/day, which was defined by the EFSA NDA panel as the lowest observed adverse effect level (LOAEL). The EFSA NDA panel recently concluded that the safety of CBD as a novel food cannot be assessed, resulting in a so-called clock stop for the applications until the applicants provide the required data. The authors suggest that certain CBD products still available on the EU market as food supplements despite the lack of authorisation should be considered "unsafe". Products exceeding a health-based guidance value (HBGV) of 10 mg/day should be considered "unfit for consumption" (Article 14(1) and (2)(b) of Regulation No. 178/2002), while those exceeding the human LOAEL should be considered "injurious to health" (Article 14(1) and (2)(a) of Regulation No 178/2002).

**Keywords:** food safety; novel foods; risk assessment; *Cannabis sativa*; tetrahydrocannabinol; food supplements; cannabidiol; benchmark dose; health-based guidance value (HBGV); liver toxicity

## 1. Introduction

In the European Union (EU), the European Food Safety Authority (EFSA) is responsible for assessing novel food applications and providing a risk assessment of the novel food, which is used by the European Commission (EC) to decide whether to authorise the product [1]. The novelty of a food is determined by the absence of a significant history of consumption prior to 15 May 1997 [2]. Extracts and derived products containing cannabinoids, such as cannabidiol (CBD), as well as isolated or fully synthetic cannabinoids, are considered novel foods [3]. Therefore, CBD products intended to be marketed as food or food supplements in the EU require prior authorisation. Despite being widely advertised and sold in increasing quantities, all CBD oils and CBD-containing food supplements available in the EU are currently marketed in violation of food laws [4]. This is no longer a

niche, as the total 2020 EU CBD market has been estimated at EUR 1.6 billion [5]. It appears to be a global phenomenon that illegality is not a deterrent for producers, as CBD foods may be readily available in jurisdictions where they are illegal due to lax enforcement [6].

As of mid-March 2022, the industry has submitted more than 150 novel food applications for CBD products, and 19 are currently under review by the EFSA Panel on Nutrition, Novel Foods, and Food Allergens (NDA). Most of the applications are for CBD extracted from *Cannabis* plants, but there are also several applications for chemically synthesised CBD [7].

In the case of CBD, the EFSA NDA panel identified several knowledge gaps during its initial assessment of the application files that need to be addressed before the safety evaluation of the compound can be concluded. That is, the effects of CBD on the liver, gastrointestinal tract, endocrine system, nervous system, psychological function, and the reproductive system need to be clarified [7]. Of all of these, the available literature suggests that one of the most important adverse effects of CBD at therapeutic doses (typically 5–25 mg/kg bodyweight (bw)/day) appears to be liver injury, which can cause symptoms of hepatitis even in healthy adults [8]. The literature was searched and reviewed by the EFSA NDA panel, but it was not possible to identify any "no-observed adverse effect levels" (NOAELs) from the available animal and human studies [7]. The EFSA NDA panel also concluded that the safety of CBD as a novel food cannot be assessed at this time, leading to a so-called clock stop for the applications until the applicants provide the required data [7].

The aim of this article is to provide an in-depth look at the available data on CBD and to provide an assessment for risk management of products currently on the market. As NOAELs were unavailable or uninformative, benchmark dose–response modelling was performed on the data highlighted by the EFSA NDA panel to provide an alternative point of departure (POD) for toxicological risk assessment.

## 2. Materials and Methods

This article uses several methods to assess the safety of CBD as a novel food. First, an updated literature search was performed to review the available data on CBD. Second, benchmark dose (BMD)–response modelling of several bioassays was conducted to provide an alternative POD for toxicological risk assessment. Third, the suitability of the benchmark dose–response modelling was verified. Finally, the data were evaluated to provide food policy recommendations.

The data analysed in this study were based on the statement of the EFSA NDA panel [7]. An additional search on PubMed for the terms "cannabidiol" or "hemp extract" and "safety" revealed the informative study by Dziwenka et al. [9], which was not included in the EFSA statement [7].

Data were screened for suitability for benchmark dose–response modelling according to the criteria of Hindelang et al. [10]: (i) a study considered for inclusion had to have administered at least three different doses and a control group receiving vehicle, while dose spacing was not considered relevant, (ii) applied doses had to be administered in mg/kg of body weight, (iii) the number of animals per dose group had to be reported, and (iv) studies reporting concomitant treatment with other drugs or medications were not included.

The eligible studies were assessed using the benchmark dose (BMD) approach according to the guidelines of the United States (US) Environmental Protection Agency (EPA) [11]. The BMD and its respective lower confidence interval, the BMDL, were calculated by fitting multiple statistical models using the benchmark dose v. 3.2.0.1 (rel. 2022-03-15) software (BMDS) [12] (US EPA, Washington, DC, USA), which performs automated fitting of selected models to dose–response data retrieved from toxicological studies. The most appropriate model was determined based on the Akaike information criteria generated in the output. All settings of the BMDS were at default.

### 3. Results

From the studies assessed by the EFSA NDA panel [7], only three animal studies with suitable dose–response data for benchmark dose modelling were identified, and an additional study by Dziwenka et al. [9] was included. Two of the studies (GWTX1412 and GWTX1413) were published in the context of the approval process of the CBD medicinal product Epidiolex as part of the application review files on the US Food and Drug Administration (FDA) website [13]. Another study, by Marx et al. [14], was published in the peer-reviewed literature, but the test object was a *Cannabis* extract and not isolated CBD. As the extract had a comparatively high purity of CBD, the authors decided to include the study for comparative reasons. Similarly, Dziwenka et al. [9,15] recently presented two studies on *Cannabis* extracts; while the 2020 study [15] did not provide the raw data necessary for BMD modelling, the 2021 study [9] was included for comparative purposes. From the available endpoints, effects on the liver, such as centrilobular hypertrophy, hepatocyte hypertrophy, increased liver weight, or relative liver weight, were selected as endpoints for modelling.

The results of the dose–response modelling are shown in Table 1. An example of the BMD modelling of the GWTX1412 study, which was considered to be the most informative, is shown in Figure 1. The full BMD modelling reports of all studies included in Table 1 are provided in the Supplementary Materials (Documents S1–S5).

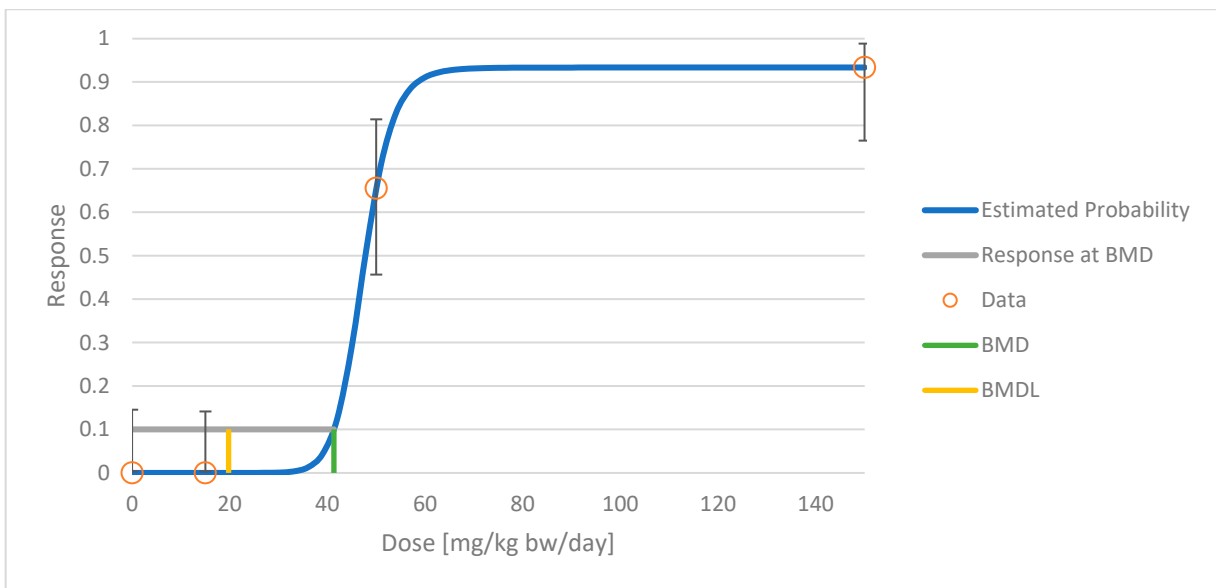

**Figure 1.** Benchmark dose (BMD) modelling of cannabidiol (CBD) for centrilobular hypertrophy of the liver in a 26-week oral study in rats (GWTX1412, see Table 1): frequentist dichotomous Hill model with benchmark response (BMR) of 10% extra risk for the BMD and 95% lower confidence limit (BMDL).

From the animal study modelling results, the authors suggest using the BMDL of 20 mg/kg bw/day from the GWTX1412 study in rats as POD, as this is the lowest—that is, the most conservative—value from the informative studies. The authors do not consider the BMDL of the GWTX1413 study to be meaningful because the dose–response model resulted in significant extrapolation beyond the lowest non-zero dose. Other studies of low-tetrahydrocannabinol (THC) *Cannabis* extracts confirmed the correctness of the order of magnitude of the GWTX1412 data because the BMDL values were quite similar considering the uncertainties of BMD modelling efforts.

**Table 1.** Results of the dose–response modelling for cannabidiol (CBD) in different animal experiments.

| Study, Animal Model | Study Design, CBD Doses | Endpoint | Sex | Model [a] | *p*-Value [b] | BMD [c] (mg/kg bw/day) | BMDL [d] (mg/kg bw/day) |
|---|---|---|---|---|---|---|---|
| GWTX1412 [13], rats | 26-week oral at doses of 0, 15, 50, and 150 mg/kg bw/day (*n* = 15/sex/group) | Liver, centrilobular hypertrophy [e] | Males + females combined [f] | Dichotomous Hill | 0.9989 | 41 | 20 |
| GWTX1413 [13], dogs | 39-week oral at doses of 0, 10, 50, and 100 mg/kg bw/day (*n* = 4/sex/group) | Liver, hepatocyte hypertrophy [e] | Males + females combined [f] | Log-Probit | 0.5771 | (3) [g] | (2) [g] |
| Marx et al. [14], rats | 90-day oral at doses of 0, 25, 90, and 180 mg/kg bw/day (*n* = 10/sex/group) [h] | Liver weight | Males [i] | Exponential 2 | 0.5235 | (52) [j] | (43) [j] |
| | | | Females [i] | Polynomial 3 | 0.9771 | (52) [j] | (34) [j] |
| Dziwenka et al. [9], rats | 90-day oral at doses of 0, 6.3, 22.7 and 81.6 mg/kg bw/day (*n* = 10/sex/group) [k] | Relative liver weight | Females | Exponential 2 | 0.1941 | (39) [j] | (26) [j] |

[a] Data from the viable recommended model selected with BMDS 3.2.0.1 (rel. 2022-03-15) software is presented. [b] A *p*-value greater than 0.1 indicates that the model fits the data (*p*-value 1.0 = perfect fit). [c] BMD: benchmark dose for a benchmark response of one standard deviation (continuous models) or 10% extra risk (dichotomous data). [d] BMDL: 95% lower one-sided confidence limit of the BMD. [e] The sum of incidences for all grades of liver effects was evaluated. [f] A single curve was fitted to both sexes as the analysis revealed no significant differences in dose–response between the sexes. [g] BMD and BMDL are both 3× lower than the lowest non-zero dose, and the model must be cautiously interpreted. [h] The study by Marx et al. [14] was conducted with a *Cannabis* extract containing 26% cannabinoids of which 96% was CBD. The dose levels were adjusted to reflect pure CBD. [i] Due to a lack of raw data, the sexes could not be combined in this case, despite no obvious differences between the sexes in this study as well. [j] Data is shown for comparative reasons only because CBD was applied in the form of a mixture with other phytochemicals from *Cannabis*. [k] The study by Dziwenka et al. [9] was carried out with a *Cannabis* oil extract containing 28.14% cannabinoids and 25.2% CBD. The levels were adjusted to reflect pure CBD. The dose–response models for males were questionable.

None of the human studies reported by the EFSA NDA panel [7] were sufficient for dose–response modelling. Therefore, the lowest observed adverse effect level (LOAEL) of 4.3 mg/kg bw/day, specifically highlighted by the EFSA panel in their presentation [16], was used as the POD. The original study from which the EFSA NDA panel derived this LOAEL was a randomized clinical trial of 120 healthy male and female healthcare professionals receiving 300 mg of CBD for 28 days. Four participants (6.8%) had elevated levels of the liver enzymes alanine aminotransferase (ALT) and aspartate aminotransferase (AST) (one critical and three mild) [17].

The PODs from animal and human data were then used to estimate health-based guidance values (HBGV) using appropriate uncertainty factors (Table 2). Overall, the authors suggest using the human HBGV of 0.14 mg/kg bw/day for risk assessment, as it is more conservative than the animal HBGV and human data should always be preferred. However, since both animal and human HBGV are in excellent agreement, the animal data provide an independent validation of the correct magnitude of the human HBGV.

**Table 2.** Calculation of health-based guidance values (HBGV) for cannabidiol (CBD) based on animal and human data.

| CBD | Animal Data | Human Data |
| --- | --- | --- |
| Type of point of departure (POD) | BMDL, see Table 1 | LOAEL [7,17] |
| Value of point of departure (POD) | 20 mg/kg bw/day (1400 mg/day [a]) | 4.3 mg/kg bw/day (300 mg/day [a]) |
| Uncertainty factor (UF) | 100 [b] | 30 [c] |
| Health-based guidance value (HBGV) | 0.20 mg/kg bw/day (14 mg/day [a]) | 0.14 mg/kg bw/day (10 mg/day [a]) |

[a] Calculation for a 70 kg human standard weight [18]. [b] Default UF of 100 (10 for inter-species variability × 10 for intra-human variability [18]). [c] Overall UF of 30 (3 for extrapolation from LOAEL to NOAEL × 10 for intra-human variability, as previously suggested by EFSA for tetrahydrocannabinol (THC) [19]).

## 4. Discussion

Despite the lack of data on CBD safety, correctly specified by the EFSA NDA panel [7] and also in a recent review by Nyland and Moyer [6], the authors believe that the available data already allow for a risk assessment if the dose–response information contained in the available data is appropriately considered. The authors also believe that the principle of precautionary public health protection demands the use of those data. In a previous article, the authors commented on the THC contamination of CBD products, stating that it is a "scandal" because unapproved and potentially unsafe products are being placed on the food market within the EU [20]. Other authors have similarly characterised the CBD market as containing "black sheep" who disregard regulations and try to make quick money with the hype surrounding cannabis legalisation [21].

The risk assessment based on available bioassays and human data on CBD toxicity proposed in this article reinforces this assessment, as many products on the market would exceed the estimated HBGV of 10 mg/day. For example, there are several CBD oil products on the market containing 10% CBD, which means that the HBGV would be contained in an amount of 0.1 g, typically found in only 3–4 drops of the product. The usually recommended dosage of several drops per day may, therefore, exceed the HBGV. For some products, which may contain even higher concentrations of CBD of up to 70%, the possible intake can exceed the LOAEL of approximately 300 mg/day (Table 3).

The HBGV of 10 mg/day proposed in this article is very similar to another approach for risk assessment by the Swiss Federal Food Safety and Veterinary Office (FSVO), which determined an oral daily dose of 12 mg CBD/adult that should not be exceeded [22]. The FSVO based its recommendation on a healthy volunteer phase I study, in which 5 out of 12 healthy subjects developed ALT elevations above the normal range at 5 mg/kg/day during the three-week treatment period [23]. The FSVO uses an uncertainty factor of 30, similar to the proposal in this study (Table 2), to calculate the guidance value.

**Table 3.** Exceedance of LOAEL in a survey of CBD food supplement products from the market in Germany (*n* = 144).

| % CBD | Number of Samples | Intake Amount [g] to Exceed LOAEL | Number of Drops to Exceed LOAEL [a] |
|---|---|---|---|
| 70 | 1 | 0.4 | 15 |
| 48 | 1 | 0.6 | 22 |
| 30 | 3 | 1.0 | 36 |
| 27 | 3 | 1.1 | 40 |
| 25 | 5 | 1.2 | 43 |
| 24 | 2 | 1.3 | 45 |
| 20 | 12 | 1.5 | 54 |
| 18 | 3 | 1.7 | 60 |
| 15 | 15 | 2.0 | 72 |
| 12 | 1 | 2.5 | 90 |
| 10 | 49 | 3.0 | 108 |
| 9 | 2 | 3.3 | 120 |
| 8 | 1 | 3.8 | 135 |
| 6 | 1 | 5.0 | 180 |
| 5 | 33 | 6.0 | 216 |
| 4 | 2 | 7.5 | 270 |
| 3 | 3 | 10.0 | 360 |
| 2.75 | 3 | 10.9 | 392 |
| 2.5 | 2 | 12.0 | 431 |
| 2 | 2 | 15.0 | 539 |

[a] Calculated with assumptions: 0.03 mL/drop; density of CBD oil 0.927 g/mL.

The liver effects that are consistently observed in all tested species, including humans, are clearly a major cause for concern. Hepatocyte hypertrophy, as observed in experimental animals, is typically associated with microsomal enzyme induction secondary to exposure to certain xenobiotics [24]. In humans, an increase in serum aminotransferases, such as ALT and AST, is clinically relevant and is usually the result of acute or chronic liver injury [25].

The primary limitation of this article is the lack of data necessary to make a definitive assessment of the safety of CBD as a novel food, specifically regarding dose–response data and clinical data in the lower concentration ranges expected in foods. It must be considered that this risk assessment concerns foods, for which safety must be generally guaranteed, unlike medicinal products, for which risk–benefit considerations must be included. For CBD-containing foods, it must also be considered that they can be consumed daily for life without medical supervision or any form of nutrivigilance, which is not mandatory in the EU.

The authors suggest that CBD products still available on the EU food market despite lack of authorisation must be assessed by risk management if they could be "unsafe" in the sense of Article 14(1) of the Basic Food Regulation No. 178/2002 [26]. If they exceed the HBGV, they would be "unfit for consumption" (Article 14(1) and (2)(b) of the Basic Regulation [26] or corresponding national regulations such as paragraph 12 of the German food and feed law). Products in exceedance of the human LOAEL of 4.3 mg/kg bw/day should be considered as being "injurious to health" (Article 14(1) and (2)(a) of the Basic Regulation [26]) and they should also be considered as being a serious risk to health in the sense of the criteria for the EU Rapid Alert System for Food and Feed (RASFF) [27], similar to the practice for THC risk assessment [28].

There is clearly a growing consumer demand for CBD and other cannabinoid products, which has not been adequately addressed by policy, leading to a huge market for unregulated CBD food supplement products, which are also marketed in supposed legal loopholes as cosmetic mouth sprays, non-food flavours, or even fantasy products for mythical animals [29]. This situation is completely unsatisfactory for consumers and industry and regulatory authorities alike. The unregulated market also leads to safety problems beyond cannabinoids, such as contamination with pesticides, heavy metals, or microbiological risks,

or even the addition of synthetic cannabinoids [6]. There is also a lack of quality control, leading to inconsistent labelling and unpredictable dosing [30]. The potential harms that CBD-containing products may pose to consumers include adverse effects of ingredients and drug interactions but also misleading practices through mislabelling or unproven or exaggerated health and disease-related claims. Risk communication could help consumers make more informed decisions about their use of CBD-containing products.

The challenges of quality control for CBD-containing products include difficulties in regulating and standardizing their production. These challenges arise due to the variability of CBD potency and purity across different products, and the lack of clear regulations and oversight in the industry. Policymakers and industry leaders must take steps to improve quality control, ensuring that consumers have access to safe and reliable products. Risk managers and other stakeholders should engage in proactive risk communication to inform consumers about potential risks and how to reduce the potential for harm.

However, as the EFSA NDA panel has pointed out, the lack of necessary data for final risk assessment means that novel food approval for CBD-containing products could still take years. This includes the time required to perform chronic toxicity studies for the missing endpoints in the low-dose range expected in foods. The authors suggest that the risk management of the European Commission and national authorities should respond with how to proceed until novel food applications are finalised. Continuing with the complete prohibition of CBD food supplements is not an effective policy to consider, as it has not worked in the past 5 years. Consumers are still ingesting CBD in considerable amounts. The authors suggest four pathways to proceed with, along with some remarks on the consequences of these risk management options:

1. Continuing with the current form of regulatory prohibition (i.e., proceeding with the current form of an unregulated market). This is probably the worst option as the prohibition approach has not worked in the past in similar areas of drug policy [31]. It is not expected that enforcement could be strengthened to the extent that CBD products could be completely removed from the market. Prohibition will only reduce consumer protection as the products will mostly drift into areas of completely unregulated and uncontrolled non-food products (such as CBD air fresheners), and effective enforcement of the novel food regulation also seems generally impossible in other areas [32].

2. Approving low-dose CBD food supplements (up to 10 mg/day and less than 300 mg/package) on an intermediary basis (see also the post-Brexit UK approach), including warning labels about potential toxic effects. This option may give consumers easy access to low-dose CBD supplements, despite the uncertain risk. However, warning labels about potential toxic effects and maximum daily doses may deter some consumers from using the products and increase the safety margin. Additionally, the lack of strict regulation may lead to variations in the quality and safety of CBD products between manufacturers.

3. Regulating low-dose CBD products as over-the-counter medicinal products available only in pharmacies, as an additional category to the already available prescription-based high-dose CBD medicinal products (see the suggestion by Health Canada [33]). This option may ensure that consumers have access to safe and high-quality CBD products. However, limiting the availability of low-dose CBD products to pharmacies may reduce their accessibility to consumers, and hence still allow for unregulated markets specifically on the internet.

4. Regulating CBD products outside the scope of foods or medicines in a separate framework, e.g., within the currently planned controlled distribution of cannabis to adults for recreational use in licensed stores in Germany. This option may ensure that CBD products are subject to strict regulation, labelling, and quality control standards. Additionally, it may provide consumers with accessibility to CBD products only through licensed stores. However, the potential for more or less unrestricted use of CBD products for recreational purposes may arise, leading to public health concerns.

Each risk management option has its advantages and disadvantages. The choice of the option will depend on the objectives of regulators and the needs of consumers. Ultimately, a comprehensive regulatory framework that ensures the safety, quality, labelling and appropriate accessibility of CBD products, possibly including some form of medical supervision such as the monitoring of liver function, is necessary to protect consumers and promote public health. The decision on how to regulate CBD products is now a political one, and the authors hope that the legislators will not turn a blind eye to the problem as they have in the past.

**5. Conclusions**

This article provides an in-depth analysis of the available toxicological data on CBD, including benchmark dose–response modelling on data highlighted by the EFSA NDA panel, and an additional informative study not included in the EFSA statement. The article also examines the suitability of benchmark dose–response modelling for providing a more comprehensive assessment of the safety of CBD as a novel food.

The main conclusions of this article are as follows:

- The currently available data for CBD do not allow a conclusive assessment of its safety as a novel food.
- Benchmark dose (BMD) response modelling of several bioassays provides an alternative POD for toxicological risk assessment.
- The authors suggest that certain CBD products still available on the EU market as food supplements despite the lack of authorisation must be considered "unsafe": Products exceeding a health-based guidance value of 10 mg/day must be considered "unfit for consumption", while those exceeding the human LOAEL must be considered "injurious to health".
- This risk assessment could only be superseded if further dose–response data become available, such as those expected from the novel food applicants.
- By encouraging risk managers to take a more active role in communicating about CBD-containing products, public health outcomes could be improved and the potential for harm reduced.

**Supplementary Materials:** The following supporting information can be downloaded at: https://www.mdpi.com/article/10.3390/psychoactives2010005/s1, BMDS 3.0 analysis reports: Document S1: GWTX1412; Document S2: GWTX1413; Document S3: Marx et al., 2018 [14] (males); Document S4: Marx et al., 2018 (females); Document S5: Dziwenka et al., 2021 [9].

**Author Contributions:** Conceptualization, D.W.L. and C.S.; methodology, D.W.L.; software, D.W.L.; validation, D.W.L.; formal analysis, D.W.L.; investigation, D.W.L.; resources, S.G.W.; data curation, D.W.L.; writing—original draft preparation, D.W.L.; writing—review and editing, C.S. and S.G.W.; visualization, D.W.L.; supervision, D.W.L.; project administration, D.W.L.; funding acquisition, S.G.W. All authors have read and agreed to the published version of the manuscript.

**Funding:** This research received no external funding.

**Institutional Review Board Statement:** Not applicable.

**Informed Consent Statement:** Not applicable.

**Data Availability Statement:** Publicly available datasets were analysed in this study. This data can be found here: https://www.accessdata.fda.gov/drugsatfda_docs/nda/2018/210365Orig1s000 PharmR.pdf (accessed 29 July 2022).

**Acknowledgments:** Presented at the 3rd International Electronic Conference on Foods: Food, Microbiome, and Health—A Celebration of the 10th Anniversary of Foods' Impact on Our Wellbeing, online, 1–15 Oct 2022 (presentation doi:10.3390/Foods2022-13022). Janin Gerstenlauer and Tabea Dietz are thanked for help in the retrieval of FDA data on cannabidiol. The graphical abstract was generated by AI using the phrase "a bottle of cannabidiol oil and a hemp leaf in front of a stop sign" using DALL-E 2 on OpenAI.com.

**Conflicts of Interest:** The authors declare no conflict of interest.

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
