# Peer review of "Does Cannabidiol (CBD) in Food Supplements Pose a Serious Health Risk? Consequences of the European Food Safety Authority (EFSA) Clock Stop Regarding Novel Food Authorisation"

_psychoactives, doi:10.3390/psychoactives2010005_

Round 1

Reviewer 1 Report

The article is based on a legitimate public health concern, already expressed in previous articles from the authors. It proposes health-based guidance values for risk management of CBD-containing products until EFSA opinion on CBD risk assessment is available. Authors present an approach that in some extent integrate risk assessment and risk management, whereas EU option pertaining to food regulation is precisely to keep them separate. Authors suggest several risk management options without discussing their respective consequences, thus weakening the overall contribution of the study to a potential improvement of the situation posed by these CBD-containing products.

Scientific approach of the study is sound but authors could have built on their experience in the field to further develope the possible consequences of such an intermediate “preliminary risk assessment“ option in the context of the chaotic marketing of CBD products.

l 38: ref 1 can be considered as questionable, as it is a very specific paper from the authors used to illustrate a very general statement.

l 39: “the hemp plant Cannabis sativa“ : hemp comprises some chemotypes/varieties of C. sativa only, thus this sentence may be considered as misleading. Authors could rephrase, e.g. “regarding hemp, i.e. the authorized varieties of C. sativa...”

l 41 : "not novel” : authors could cite the NF catalog as reference.

l 60 : authors base their “preliminary assessment” on liver toxicity only. This could be more emphasized in the paragraph dealing with limitations of the study at the end of the discussion section.

l 61: “therapeutic doses”: it could be informative to give a range of doses here, and to further explicit the meaning of “therapeutic” here (e.g. the doses used in clinical trials evaluating efficacy of CBD against diseases).

l 63-63: “no observed adverse effect level could not be identified" : please check syntax.

l 64 : it looks like the following sentence would be more appropriate: "EFSA NDA Panel concluded that the safety of CBD as a novel food cannot currently be evaluated”

Material and methods: a description of how the literature search update was performed would be appreciated (keywords, literature database, etc.).

l 170 : authors mentions the concept of “preliminary risk assessment“. Authors could put the relevance of such a concept in perspective with the current regulatory frame.

General comment on the Discussion section content: Authors propose "meanwhile" risk management options for the sake of public health. In order to provide a balanced view on one of the option put forward by the authors (authorizing products providing < 10mg CBD/day), authors should further discuss the drawbacks of establishing such a "temporary" risk management option, subject to 1) further changes in the future and 2) appeal against a unusual, partial risk assessment in the frame of a quite defensive market. Authors exclude the “complete prohibition“ option based on its failure so far, without analyzing the peculiar context around CBD and cannabis that led to such a situation. Such an analysis would be useful to foresee the relevance and applicability of any new risk management option, including those proposed by the authors. The two other options proposed by the authors imply a dedicated regulatory framework to these products, and the consequences of such a regulatory niche are not discussed.

Authors could also further elaborate on the usefulness of the study to (again) 1) illustrate the current risk of CBD-containing products, 2) discuss the challenges of quality control of CBD-containing products and 3) incite risk managers to communicate on the risk associated to the consumption of such products.

Author Response

The article is based on a legitimate public health concern, already expressed in previous articles from the authors. It proposes health-based guidance values for risk management of CBD-containing products until EFSA opinion on CBD risk assessment is available. Authors present an approach that in some extent integrate risk assessment and risk management, whereas EU option pertaining to food regulation is precisely to keep them separate. Authors suggest several risk management options without discussing their respective consequences, thus weakening the overall contribution of the study to a potential improvement of the situation posed by these CBD-containing products.

RESPONSE: Thank you for your assessment of our paper. We agree with your comments on risk assessment and risk management. Unfortunately, risk management in the EU has so far turned a blind eye to CBD and basically left local authorities without guidance. Risk assessment has not been very informative either. For example, the EFSA document did not even propose a health-based guidance value. As both aspects of the holistic risk assessment are basically insufficient at the moment, we have decided, in our capacity as a state institute tasked with providing expert opinions on cannabis products, to fill the gap and provide an interim risk assessment until the EU authorities complete their tasks. The EU's Rapid Alert System for Food and Feed shows that apparently several food control authorities are following our example and notifying some cannabidiol products as "serious health risk". In addition, in response to your comment about the weak discussion of the implications of the risk management option, we have improved the discussion on page 7.

Scientific approach of the study is sound but authors could have built on their experience in the field to further develope the possible consequences of such an intermediate “preliminary risk assessment“ option in the context of the chaotic marketing of CBD products.

RESPONSE: Thank you for your suggestion. We have significantly expanded the discussion of risk assessment. See also previous response.

l 38: ref 1 can be considered as questionable, as it is a very specific paper from the authors used to illustrate a very general statement.

RESPONSE: Although not relevant to the present case, the statements made in L 38 are in fact supported by the information contained in Ref. 1.

l 39: “the hemp plant Cannabis sativa“ : hemp comprises some chemotypes/varieties of C. sativa only, thus this sentence may be considered as misleading. Authors could rephrase, e.g. “regarding hemp, i.e. the authorized varieties of C. sativa...”

RESPONSE: We do not fully agree and this may be a matter of language. In German usage, hemp (or German "Hanf") is used more or less synonymously with the term cannabis, and does not restrict it to specific species and varieties. Probably, especially in the USA, hemp can only be used for low-THC cannabis, while in Europe it might be called "industrial hemp" or "fibre hemp", as opposed to "drug or medicinal hemp". As hemp seeds are not discussed in the rest of the text, we decided to remove this sentence to avoid confusion. We have also removed most of the word "hemp" throughout the article to avoid any confusion in terminology.

l 41 : "not novel” : authors could cite the NF catalog as reference.

RESPONSE: The statement has been removed in response to the previous comment.

l 60 : authors base their “preliminary assessment” on liver toxicity only. This could be more emphasized in the paragraph dealing with limitations of the study at the end of the discussion section.

RESPONSE: This is correct and we have included it in the limitations section. Nevertheless, the dose-response data and the human data show that liver toxicity should be the most sensitive endpoint.

l 61: “therapeutic doses”: it could be informative to give a range of doses here, and to further explicit the meaning of “therapeutic” here (e.g. the doses used in clinical trials evaluating efficacy of CBD against diseases).

RESPONSE: The therapeutic dose range for Epidiolex, a CBD medicine approved in the EU and US, is 5-25 mg/kg bw/day (https://www.epidiolexhcp.com/dosing-and-calculator).

l 63-63: “no observed adverse effect level could not be identified" : please check syntax.

RESPONSE: The sentence was correct in principle, but we have revised it for clarity.

l 64 : it looks like the following sentence would be more appropriate: "EFSA NDA Panel concluded that the safety of CBD as a novel food cannot currently be evaluated”

RESPONSE: The sentence has been changed as requested.

Material and methods: a description of how the literature search update was performed would be appreciated (keywords, literature database, etc.).

REPONSE: Search terms and database added.

l 170 : authors mentions the concept of “preliminary risk assessment“. Authors could put the relevance of such a concept in perspective with the current regulatory frame.

RESPONSE: The reviewer is correct that the term "preliminary" may not be entirely appropriate as any risk assessment is based on the available data and may be preliminary in the sense that it may be superseded as further data become available. In addition, there is already an administrative court decision in Germany on our risk assessment, which is supported by the court decision (VG Karlsruhe, decision of 24.2.2023 Az. 3 K 515/23). We have therefore removed the word 'preliminary' throughout the document.

General comment on the Discussion section content: Authors propose "meanwhile" risk management options for the sake of public health. In order to provide a balanced view on one of the option put forward by the authors (authorizing products providing < 10mg CBD/day), authors should further discuss the drawbacks of establishing such a "temporary" risk management option, subject to 1) further changes in the future and 2) appeal against a unusual, partial risk assessment in the frame of a quite defensive market. Authors exclude the “complete prohibition“ option based on its failure so far, without analyzing the peculiar context around CBD and cannabis that led to such a situation. Such an analysis would be useful to foresee the relevance and applicability of any new risk management option, including those proposed by the authors. The two other options proposed by the authors imply a dedicated regulatory framework to these products, and the consequences of such a regulatory niche are not discussed.

RESPONSE: Prohibition is usually the least desirable regulatory option, as it opens the field to unregulated black or grey markets, which is exactly what we have now with CBD products. However, the reviewer is right that continuing with prohibition is certainly an option. We have included it as a fourth option in the discussion. Apart from that, the analysis of the market situation and the regulatory framework is not the core competence of the author team. The policy analysis of CBD regulation would probably be the subject of a separate full paper.

Authors could also further elaborate on the usefulness of the study to (again) 1) illustrate the current risk of CBD-containing products, 2) discuss the challenges of quality control of CBD-containing products and 3) incite risk managers to communicate on the risk associated to the consumption of such products.

RESPONSE: Thank you for your suggestions. We have included the options in the discussion. The last option in particular is important. None of the CBD manufacturers currently warn consumers about liver risks.

Reviewer 2 Report

The authors have evaluated the additional data regarding the safety of CBD products and substantially contributed to health risk assessment. I suggest no corrections

Author Response

Thank you for the assessment of our paper!

Reviewer 3 Report

The authors discuss very attractive topic since there is a growing interest in CBD-containing food. This manuscript is important from both scientific as well as the legislative view. The manuscript is well organized, results are clearly presented and references are up-to-date. Conclusions are supported by the results. I recommend acceptance of the paper in the present state.

Author Response

Thank you very much for reviewing our paper!

Reviewer 4 Report

The manuscript submitted for review is very well structured. The criteria for inclusion of data in the study were described in detail. The manuscript contains information from both regulatory and scientific publications. The limitation of the work, which is the small amount of data, is mentioned in the final part of the discussion. The publication reads very well and the subject matter discussed in it refers to a very important and current problem.

Author Response

Thank you for the assessment of our paper and stressing the limitations of the data. Considering reviewer #1, we have additionally improved the discussion.

Round 2

Reviewer 1 Report

The revised manuscript is satisfactory, thank you.